



# An analytical formulation for turbulent kinetic energy added by wind turbines based on large-eddy simulation

Ali Khanjari[1], Asim Feroz[1], and Cristina Archer[1]

[1]Center for Research in Wind (CReW), University of Delaware, 221 Academy St, Newark, DE 19716, USA

**Correspondence:** Cristina Archer (carcher@udel.edu)

**Abstract.** Wind turbine wakes are plume-like regions characterized by reduced wind speed and enhanced turbulent kinetic energy (TKE) that form downstream of wind turbines. Numerical mesoscale models, like the Weather Research and Forecast (WRF) model, are generally effective at reproducing the wind speed deficit, but lack skills at simulating the TKE added by wind turbines. Here we propose an analytical formulation for added TKE by a wind turbine that reproduces, via least-square error parameter fitting, the main features of the three-dimensional structure of added TKE as simulated in previous large-eddy simulation (LES) studies, including: a streamwise peak at $x$ = 4–6D (where D is the turbine diameter), a vertical peak near the upper rotor region, and an annular Gaussian-like distribution along the rotor edge. Validation of the proposed formulation against independent LES results and wind tunnel observations from the literature indicates a promising performance in the case of a single wind turbine wake. The ultimate goal is to insert the proposed formulation, after further improvements, in the WRF model for use within existing or new wind farm parameterizations.

## 1 Introduction

Over the last two decades, there has been a significant surge in renewable energy, especially wind energy, due to its vast potential worldwide (Archer and Jacobson, 2005) and to the global shift towards low- or no-carbon sources of energy to fight global climate change. For the first time in history wind has provided over 10% of the total electricity production of the world, with a global total installed capacity of approximately 824 GW (International Renewable Energy Agency (IRENA), 2022).

When wind turbines are installed in a wind farm, their layout is a compromise between two opposite needs. On one hand, the distance between turbines should be as small as possible to minimize the total lease area and to reduce cable length costs. On the other hand, their distance should be as large as possible to minimize so-called "wake losses" and maximize total power output. Wake losses are the undesirable consequence of wakes, which are plume-like regions characterized by reduced wind speed and enhanced turbulent kinetic energy (TKE) that form downstream of wind turbines (Stevens and Meneveau, 2017). As a result of the reduced wind speed and the enhanced turbulence in the wakes, not only is the performance of downstream turbines in the same wind farm reduced (Archer et al., 2018), but so is also the power production of neighbouring wind farms (Nygaard, 2014). As such, wake losses are one of the most important issues affecting wind farm production today (Göçmen et al., 2016) and the correct estimation and prediction of wind turbine wakes and their evolution are fundamental to ultimately ensure reliable power production from a wind farm (Ye et al., 2023).



Historically, most analytical wake models were developed to address the issue of wake impacts on power production of downstream turbines, thus they focused on the most relevant parameter, the wind speed deficit; only a few were developed to address the issue of added turbulence (discussed later). Jensen (1983) proposed the first analytical wake model to predict wind speed deficit behind a wind turbine, based on conservation of momentum. The Jensen model predicts a uniform top-hat wind speed deficit across an expanding rotor area that grows with downstream distance $x$ at a specified constant rate. Katic et al. (1986) proposed a refinement to the Jensen model by introducing an axial induction factor and a method to account for overlapping wakes. Larsen (1988) developed a semi-analytical model from asymptotic expressions of Prandtl's rotational symmetric turbulent boundary layer equations. By fitting data from field measurements, Barthelmie et al. (2004) proposed an analytical formula for the hub-height wind speed deficit as an inverse power function of $x$. Frandsen et al. (2006) introduced an analytical wake model aimed at predicting the wind speed deficit in large offshore wind farms with a rectangular layout and equal spacing between the wind turbines, based on conservation of mass and momentum.

The first Gaussian model for the wind speed deficit was developed by Bastankhah and Porté-Agel (2014), who assumed that the self-similar wake would expand at the same rate in the vertical direction z as along the spanwise direction y. Xie and Archer (2015) improved the Gaussian wake model by using two different expansion rates along $z$ and $y$, the values of which were found by fitting high-fidelity simulation results. Gao et al. (2016) proposed a two-dimensional version of the Jensen wake model using the Gaussian function to improve the prediction of the velocity deficit at each cross-section downstream. Wang et al. (2023) proposed a modification to the Gaussian analytical wake model by considering an empirical error term to eliminate the gap between measured and calculated values. However, this Gaussian analytical wake model is only two-dimensional in the lateral-longitudinal ($x - -y$) plane and does not take into account the effect of wind shear and the anisotropy of turbulence intensity on the other planes. A review of the performance of analytical wake loss models for wind speed deficit caused by wind turbines can be found in Archer et al. (2018).

Only a few analytical wake models provide estimates of the added turbulence by wind turbines and they all use turbulence intensity (TI) as the variable of interest. Quarton and Ainslie (1990) suggested the first empirical formulation for the maximum added turbulence intensity by a wind turbine ($\Delta TI_{max}$) by considering the effect of freestream turbulence intensity ($TI_\infty$) and the thrust coefficient of the wind turbine ($C_T$):

$$\Delta TI_{max} = 4.8 C_T^{0.7} TI_\infty^{0.68} \left( \frac{x}{x_N} \right)^{-0.57}, \tag{1}$$

where $x_N$ is a scaling parameter that represents the length of the near wake. Several modifications to this formulation have been proposed in the literature, for example by Hassan (1993) and Xie and Archer (2015).

One of the most successful variations of Eq. 1 was that introduced by Crespo and Hernández (1996), who divided the wake into two different regions: the near-wake ($x < 3D$, where $D$ is the rotor diameter) and the far-wake ($x \geq 3D$); they then developed a different equation for each:

$$\Delta TI_{max} = \begin{cases} 0.362(1 - \sqrt{1 - C_T}) & x < 3D \\ 0.73 \left( \frac{1 - \sqrt{1 - C_T}}{2} \right)^{0.8325} (TI_\infty)^{-0.0325} \left( \frac{x}{D} \right)^{-0.32} & x \geq 3D. \end{cases} \tag{2}$$





The reason for the two different formulations in the two regions is that, in the near-wake, the influence of the rotor aerodynamics, such as blade aerodynamics, stalled flow, and the presence of tip vortices, are predominant on the wake (Vermeer et al., 2003). Tip vortices originate from the blade tips and roots, propagating downstream in helical trajectories over a short distance (Sherry et al., 2013). If the inclination angle is minimal, these tip vortices will be interpreted as cylindrical shear layers (Crespo et al., 1999). These layers expand within the wake due to turbulent diffusion, forming a ring-shaped region characterized by high turbulence intensity and substantial velocity gradients. However, the tip vortices will break down because of instability within a short distance downstream. In the far-wake region, the rotor effects are less important. The wake is fully developed and turbulent diffusion of momentum becomes dominant, the flow has less energy, and an increased level of turbulence is found, fueled by shear production. It is generally assumed that velocity and turbulence intensity should remain self-similar and axi-symmetric in the absence of ambient wind shear (Johansson et al., 2003; Xie and Archer, 2015; Cafiero et al., 2020; van der Laan et al., 2023).

The study by Ishihara and Qian (2018) hereafter referred to as IQ2018, is the first in the literature to provide a three-dimensional analytical equation for the turbulence intensity added by a wind turbine in the downstream wake. Their empirical formula contains a Gaussian function in the radial direction $r$ (from the center of the rotor) multiplied by an amplitude function (an inverse function of $x$) as follows:

$$\Delta TI(x,y,z) = \frac{1}{d + e\frac{x}{D} + f\left(1 + \frac{x}{D}\right)^{-2}} \times \left\{ k_1 \exp\left[-\frac{\left(r - \frac{D}{2}\right)^2}{2\sigma^2}\right] + k_2 \exp\left[-\frac{\left(r + \frac{D}{2}\right)^2}{2\sigma^2}\right] \right\} - \delta(z) \tag{3}$$

$$\delta(z) = \begin{cases} 0 & z \geq H \\ TI_\infty \sin\left(\pi\frac{H - z}{H}\right)^2 & z < H, \end{cases} \tag{4}$$

where $H$ is the turbine hub height, $d, e, f, k_1, k_2$, and $\sigma$ are functions of $C_T$ and $TI_\infty$ of the form $aC_T^b TI_\infty^c$. We note that also Eqs. 1 and 2 can be reduced to this same form.

Li et al. (2022) later hypothesized that the added turbulence intensity $\Delta TI$ in the streamwise direction has a similar self-similarity property as the velocity deficit and proposed a three-dimensional analytical formula for added turbulence intensity similar to that by Ishihara and Qian (2018). Tian et al. (2022) developed a three-dimensional cosine-shape model to estimate the wake turbulence intensity; they assumed that the wake has a similar growth rate in the spanwise and vertical directions and that the maximum added turbulence intensity is redistributed along the radial direction with a dual-cosine shape function.

Wind turbine wakes have also been studied with numerical wake models. The first numerical wake model was proposed by Lissaman (1979), who pre-dated Jensen (1983) and developed a numerical program to solve the complex problem of overlapping wakes in an array of multiple wind turbines. Lissaman (1979) was the first to recognize the importance of ambient turbulence in overlapping wakes, which, he stated, may have a greater impact than that due to the momentum deficit generated by the individual turbines. Over ten years later, Ainslie (1988) proposed a numerical solution for the wake development by simplifying the Navier-Stokes equations for the turbulent boundary layer and introduced ambient turbulence and its effect on





the wake decay. More recently, Yang et al. (2015) proposed a modeling framework of $\Delta TKE$ by a wind turbine in vertical planes downwind as a function of inlet velocity at hub height and thrust coefficient, which can be used to estimate the TKE of

turbine wakes in complex terrain.

Whether purely analytical or numerical, whether predicting wind speed deficit or added turbulence intensity or both, wake models are useful to understand the behaviour of a wind turbine wake, but they cannot provide any information on the effects of the wake on the surrounding environment, such as changes in vertical mixing, or surface temperature, or heat and momentum fluxes at the surface. Large-eddy simulation (LES) has been a successful numerical approach to study wind turbine wakes

(Breton et al., 2017) and their effects on the surrounding environment (Wu et al., 2023) because of their high spatial and temporal resolutions (order of a few meters and tens of seconds, respectively) and the accuracy of the actuator disk (Sørensen and Myken, 1992; Madsen, 1996; Mikkelsen, 2003) and actuator line (Sorensen and Shen, 2002) models used to incorporate the effects of the rotating blades. Many LES studies have been conducted to capture wind speed and TKE properties in wind turbine wakes (Eriksson et al., 2015; Vanderwende et al., 2016; Lee and Lundquist, 2017; Deskos et al., 2019; Siedersleben

et al., 2020; Feng et al., 2022). Notably, Wu et al. (2023) conducted LES that included the effect of atmospheric stability to show that the wind speed deficit behaves differently from the $\Delta TKE$ and that the two are not co-located in the wake region. However, LES are computationally demanding and therefore are not used for medium- or long-term wind farm power predictions, but rather for temporal horizons of the order of a few hours to a day.

Numerical weather prediction (or mesoscale) models, like the Weather Research and Forecast (WRF) (Skamarock et al.,

2021), are the preferred tool to predict weather over longer temporal horizons, from several days to several years. However, due to the coarser spatial resolution than that of LES, ranging between 1 to 100 km, numerical weather prediction models cannot resolve the details of the wind turbine wakes, which therefore need to be "parameterized." A parameterization is a way to include the effects of a process of interest that cannot be resolved directly by the numerical model, typically because the spatial resolution of the numerical model is not fine enough to explicitly treat that process. A parameterization is basically a

model-within-a-model that uses the resolved variables at each grid cell to calculate the effects of the process of interest on the resolved variables in that cell (but not the process itself). In the WRF model, several processes are parameterized, including convection, boundary layer turbulence, radiation, to name a few. The wind farm parameterization (WFP) available by default in WRF is that by Fitch et al. (2012), which treats the wind turbines in a grid cell as sinks of momentum and sources of TKE. As shown in the literature (Pan and Archer, 2018; Archer et al., 2019, 2020; Fischereit et al., 2022), the Fitch parameterization

ignores wake effects within a grid cell and treats $\Delta TKE$ in an overly simplistic way.

In summary, most studies in the literature have focused on predicting the velocity deficit caused by wind turbines; far fewer have focused on added turbulence. In the far-wake region, which is the most relevant portion of the wake for long-term impacts on the environment, TKE is formed due to the increased shear caused by the wind speed deficit in the upper part of the rotor area. If $\Delta TKE$ is not accounted for properly, inaccurate predictions of the turbulent fluxes of heat and momentum near the

surface may occur, which ultimately may cause inaccurate predictions of near-surface temperature and moisture. Here, we aim at developing an analytical formulation for $\Delta TKE$ by wind turbines that is designed to be ultimately incorporated in a future wind farm parameterization for the WRF and other numerical weather prediction models. With the understanding that any



parameterization is, by definition, an approximation, our goal in this paper is to propose a reasonable analytical formulation for added TKE that avoids over-prediction. Avoiding over-prediction of added TKE is crucial for a (future) parameterization

because a numerical mesoscale model, like the WRF, will add some TKE on its own due to the resolved vertical shear. If the parameterization over-estimates TKE in addition to the TKE added from the resolved shear, then an excess of TKE will occur. This issue of potential "double counting" of TKE was first addressed by Ma et al. (2022b, a). Our formulation for $\Delta TKE$ is inspired by that proposed by Ishihara and Qian (2018), as their analytical formula for added turbulence intensity by a wind turbine depends on atmospheric stability (via ambient turbulent intensity) and turbine technical specifications (via the thrust

coefficient).

## 2 Methods

TKE is the kinetic energy per unit mass associated with eddies in a turbulent flow, defined as half of the sum of the variances (squares of standard deviations) of the three velocity components $u, v, and w$ (along $x, y$, and $z$, respectively) as follows:

$$TKE = \frac{1}{2}\left(\overline{u'^2} + \overline{v'^2} + \overline{w'^2}\right) = \frac{1}{2}\left(\sigma_u^2 + \sigma_v^2 + \sigma_w^2\right), \tag{5}$$

where the fluctuating velocity components are the difference between the instantaneous and the average velocity component along each axis (e.g., $u' = u - \bar{u}$).

Turbulence intensities are defined along each direction as follows (Arya, 2001; Burton et al., 2011):

$$TI_x = \frac{\sqrt{\overline{u'^2}}}{\overline{U}} = \frac{\sigma_u}{\overline{U}}, \quad TI_y = \frac{\sqrt{\overline{v'^2}}}{\overline{U}} = \frac{\sigma_v}{\overline{U}}, \quad TI_z = \frac{\sqrt{\overline{w'^2}}}{\overline{U}} = \frac{\sigma_w}{\overline{U}}, \tag{6}$$

where $\overline{U}$ is the mean wind speed, generally taken at hub height and often only horizontal. If turbulence was truly isotropic,

then the three standard deviations should be approximately equal to one another. In the real atmosphere, however, where the x, y, and z axes are aligned with the west-east, south-north, and bottom-up directions, respectively, this is not true. Typically the largest one is $\sigma_u$, followed by $\sigma_v$ (approximately $0.75\sigma_u$ in neutral conditions) and then by $\sigma_w$ (approximately $0.52\sigma_u$ in neutral conditions) (Arya, 2001).

In the wind energy community, since wind turbines are generally yawed to face the mean wind, the $x$ direction is set to coin-

cide with the longitudinal (or streamwise) direction (which is not necessarily aligned west-east), and the $y$ and $z$ directions are denoted as lateral and upward. The International Electrotechnical Commission (IEC) standard for wind turbines (International Electrotechnical Commission, 2019) defines turbulence intensity as "the ratio of the wind speed standard deviation to the mean wind speed," but it effectively considers only $\sigma_u$ (referred to as $\sigma_1$ or "turbulence standard deviation", where 1 is the index for the x-axis) in its definition of TI. However, the IEC standard recognizes that the three standard deviations in Eq. 6 should

be different from one another and recommends that any wind velocity field for turbulence models used for standard turbine classes satisfy the following conditions: $\sigma_v \geq 0.7\sigma_u$ and $\sigma_w \geq 0.5\sigma_u$.

There is not a straightforward relationship between $\sigma_U$ and the standard deviations of the individual wind components $\sigma_u$, $\sigma_v$, and $\sigma_w$ because of the non-linear relationship $U = \sqrt{u^2 + v^2 + w^2}$ (thus $\overline{U} \neq \sqrt{\bar{u}^2 + \bar{v}^2 + \bar{w}^2}$ and $\sigma_U^2 \neq \sigma_u^2 + \sigma_v^2$). It can be





shown that, to a first approximation, the standard deviation of the wind speed $\sigma_U$ is close to that of the streamwise component
$\sigma_u$ (Larsén, 2022), which supports the convention used by the IEC. In light of these considerations, here we define TI as the
ratio of the wind speed standard deviation over the mean wind speed at hub height, but use the following approximations:

$$TI = \frac{\sigma_U}{\overline{U}} \approx \frac{\sigma_u}{\overline{U}} \approx \frac{\sqrt{\frac{2}{3}TKE}}{\overline{U}} \qquad (7)$$

Traditionally, only the horizontal wind speed is used for $U$ and $\overline{U}$, possibly because only the horizontal components of wind
velocity contribute to the rotation of the wind turbine blades. The approximate relationship between TI and TKE in Eq. 7
(Wilcox, 2006), which is based on the assumption of isotropic turbulence and is therefore likely to overestimate TI, is used to
convert between TI and TKE when needed (e.g., in the Validation Section 3.2).

A notable difference between Eqns. 5 and 7 is that TKE includes the vertical fluctuations of the wind field, which makes
TKE more suitable than TI for applications where vertical mixing is important, like wind turbine wake effects. In addition,
in mesoscale models like the WRF, TKE is often a prognostic variable that is directly simulated, whereas neither TI nor the
standard deviations of the wind components are (i.e., there is no equation to obtain directly the individual components of
the Reynolds stress tensor). In the field, however, it has been common to compute TI from measurements because TI can be
calculated from simple and relatively inexpensive two-dimensional cup anemometers (thus only the horizontal components
are considered), whereas sophisticated and expensive three-dimensional sonic anemometers are necessary to measure all three
components of the wind in order to derive TKE.

In the rest of this study, $x$ is the downstream distance from the wind turbine, $z$ is vertical distance from the ground, and $y$ is
lateral distance from the wind turbine (positive to the left of the turbine, facing the turbine).

## 2.1 Proposed formulation

The equation for added TI by Ishihara and Qian (2018) is the starting point of the proposed formulation because it captures
well the annular distribution of turbine-induced turbulence and its evolution into a single-peak Gaussian with distance. A few
features, however, are not well resolved: the peak of added turbulence at hub height occurs at about 1D from the turbine
position, rather than at 4-6D where shear production is highest; the peak in the vertical at the rotor top is too strong; and a
spurious peak forms below the rotor. Inspired by the formulation of Ishihara and Qian (2018), but with the intent of improving
upon the issues above, here we propose that $\Delta TKE$, normalized by the square of the upstream undisturbed hub-height wind
speed $U_\infty$, can be modeled as the product of three functions: a streamwise function $A(x)$, a radial function $G(r)$, and a vertical
function $W(z)$:

$$\frac{\overline{\Delta TKE}}{U_\infty^2} = \alpha \times A(x) \times G(r) \times W(z). \qquad (8)$$

We note that the formulation by Ishihara and Qian (2018) was similar, except it did not include a $W(z)$ function. The scalar $\alpha$
is a tuning parameter that ensures that the amplitude of $\Delta TKE/U_\infty^2$ in the wake is of the right magnitude (i.e., matches the
LES data, as described later).





The streamwise function $A(x)$ should not be exponentially decreasing, as often assumed for turbulence intensity (Quarton and Ainslie, 1990; Crespo and Hernández, 1996; Xie and Archer, 2015; Ishihara and Qian, 2018), because $\Delta TKE$ is well-known to peak at a distance $x_{max}$ between 4D and 8D from the turbine's streamwise location $x_0$ (Xie and Archer, 2015; Wu et al., 2023), not at $x_0$ (Fig. 1a). Here we propose a Weibull-like distribution for $A(x)$ as follows:

$$A(x) = \left(\frac{x - x_0}{\lambda_A}\right)^{k_A - 1} exp\left[-\left(\frac{x - x_0}{\lambda_A}\right)^{k_A}\right],\qquad(9)$$

where $\lambda_A$ and $k_A$ are the scale and shape parameters of the Weibull distribution, with $k_A$=2. The Weibull function is chosen because it is non-symmetric and because it is one-tailed, as is the observed distribution of added TKE along $x$. An example of the evolution of $A(x)$ is shown in Fig. 1a in blue.

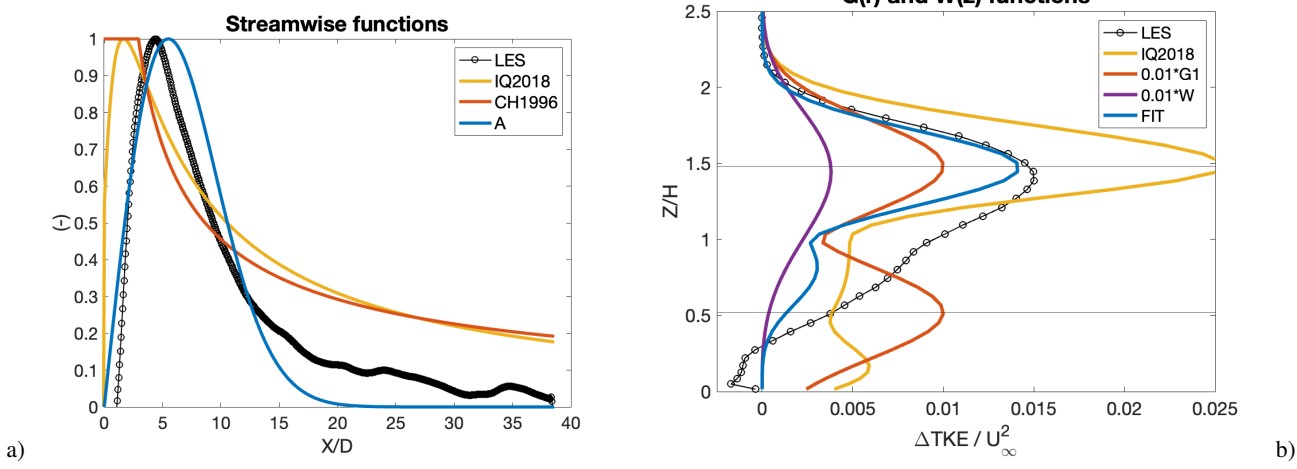

a)                                                                                                          b)

**Figure 1.** Comparison of the proposed fitting functions against the WRFLES-N results by Wu et al. (2023) (labeled 'LES'): a) streamwise function $A(x)$ at $z = H$ and $y = y_0 + D/2$ versus the fit by Ishihara and Qian (2018) ('IQ2018') and by Crespo and Hernández (1996) ('CH1996'), and b) radial and vertical functions $G(r)$ and $W(z)$ at $x = x_0 + 5D$ and $y = y_0$. Note that $G(r)$ and $W(z)$ are re-scaled for displaying purposes. The black thin lines in b) indicate the rotor top and bottom.

The radial function $G(r)$ is assumed to be a Gaussian that peaks at the tip annulus of the rotor, inspired by the $\phi(r)$ function of Ishihara and Qian (2018)Ishihara and Qian (2018), as follows:

$$G(r) = \exp\left[-\frac{(r - D/2)^2}{2\sigma_r^2}\right],\qquad(10)$$

where $r$ is:

$$r = \sqrt{(y - y_0)^2 + (z - H)^2},\qquad(11)$$

$y_0$ is the spanwise location of the turbine, and $\sigma_r$ is a linear function of $x$:

$$\sigma_r(x) = k_r(x - x_0) + \varepsilon_r D,\qquad(12)$$





where $k_r$ is the radial expansion rate (i.e., $\frac{\partial \sigma_r}{\partial x}$) and $\varepsilon_r$ is a multiplying factor to the rotor diameter that sets the initial width of
the Gaussian distribution of the added TKE along the annulus of the rotor disk.

Lastly, the vertical function $W(z)$ is also assumed to be Weibull-like:

$$W(z) = \left(\frac{z}{\lambda_W}\right)^{k_W-1} exp\left[-\left(\frac{z}{\lambda_W}\right)^{k_W}\right], \tag{13}$$

where the shape parameter $k_W$ is set equal to 4 to ensure a steeper decrease in $\Delta TKE$ above the top tip than below it. Examples
of vertical profiles of the functions $G$ and $W$ are shown in Fig. 1b with yellow and orange lines.

Due to the properties of the Weibull distribution, there is an analytical relationship between the point at which the function
reaches the maximum, which by definition is the mode ($x_m$), and the value of $\lambda$:

$$\lambda = \frac{x_m}{\left(\frac{k-1}{k}\right)^{\frac{1}{k}}}. \tag{14}$$

Thus, the values of $\lambda_A$ and $\lambda_W$ are directly related to the position of the maximum $\Delta TKE$ along $x$ and $z$, respectively. As
such, it is reasonable to expect that $\lambda_A$ and $\lambda_W$ depend on $D$ and $H$.

In summary, the equation for $\Delta TKE/U_\infty^2$ (Eq. 8) contains five unknown parameters: $\alpha, \lambda_A, \lambda_W, k_r$, and $\varepsilon_r$. Before we
explain how we obtain their values (by fitting) and why we refer to them hereafter as the five "direct-fit" parameters (in Section
2.3), we need to describe the datasets that we used for the fitting (in the next Section 2.2).

## 2.2 LES datasets

We use two independent LES datasets to calibrate the fitting parameters of our analytical model (Eqs. 8-13): the published
results of Wu et al. (2023) and Vahidi (2024), described next.

The modeling system used by Wu et al. (2023) included both an LES model (WRF-LES) (Moeng et al., 2007) and a
generalized actuator disk parameterization (WRF-LES-GAD) (Mirocha et al., 2014). A single and a row of wind turbines
were simulated under neutral, stable, and unstable conditions with different undisturbed hub-height wind speed, turbulence
intensity, and thrust coefficient (Table 1). The outer domain with horizontal grid spacing of 15 m was one-way nested with an
inner domain with finer horizontal grid spacing of 5 m (see their Figure 1). The vertical resolution started at 2.5 m near the
ground and was then stretched by 10% per level until 35 m, kept constant at 5 m from 35 m to 200 m, and finally stretched
by 5% until the model top. The lateral boundary conditions were periodic for the outer domain and time-varying for the inner
domain. The time step was 1/7 s in the outer domain and 1/21 s in the inner domain. The wind turbine was the PSU 1.5-MW
turbine with $H$ = 80 m and $D$ = 77 m, placed in the domain at 8.2D from the inlet of the inner domain and centered laterally.
The temperature profile was uniform at 297.3 K up to the initial boundary layer height (at 200 m, 500 m, and 800 m for stable,
neutral, and unstable conditions, respectively), with an inversion of 0.01 K m$^{-1}$. The desired atmospheric stability was imposed
by assigning a surface heat flux of -0.07, 0, and +0.07 K m s$^{-1}$ for stable, neutral, and unstable conditions, respectively, with
a surface roughness length $z_0$ = 0.01 m for all simulations.

The LES results of Vahidi (2024) were obtained with the WiRE-LES solver (Abkar and Porté-Agel, 2015), with a standard
actuator disk model (Meyers and Meneveau, 2010) for the axial force induced by the wind turbine. The computational domain





was 3840 m × 1920 m × 640 m with 256 × 256 × 128 grid points in the $x$, $y$, and $z$ directions, respectively, with $z_0 = 0.001$ m and no Coriolis force. The wind turbine, with $D = H = 80$ m, was located at a distance of 15D from the inlet and centered laterally. The boundary conditions in the horizontal direction were periodic for the precursor (i.e., with no wind turbine) runs.

When the turbine was inserted in the domain, an inflow boundary condition was applied to override the imposed periodic boundary condition in the streamwise direction using the prior periodic results. To smoothly adjust the flow to an undisturbed inflow condition, a buffer zone was introduced upstream of the inflow section. A constant streamwise pressure gradient was used to maintain a velocity of around 8 m s$^{-1}$ at the center of the actuator disk.

Further details about the LES suites that will be used for calibration can be found in the original studies (Wu et al., 2023;
Vahidi, 2024) and in Table 1.

**Table 1.** Simulation details and values of the five direct-fit parameters for the 17 LES cases used for the analytical model calibration. The label "WRFLES" refers to the WRF-LES dataset by Wu et al. (2023), where "S", "N", and "U" refer to stable, neutral, and unstable conditions; "VPA-TI064" and "VPA-TI107" refer to the neutral-stability simulations by Vahidi (2024), with TI set to 0.064 and 0.107.

| Calibration case | D | H | $U_{hub}$ | $C_T$ | $TI_\infty$ | $\alpha$ | $\lambda_A$ | $\lambda_W$ | $k_r$ | $\varepsilon_r$ |
|---|---|---|---|---|---|---|---|---|---|---|
| | (m) | (m) | (m s$^{-1}$) | (-) | (-) | ($\times 10^{-2}$) | (m) | (m) | ($\times 10^{-2}$) | ($\times 10^{-1}$) |
| WRFLES-S | 77 | 80 | 8.96 | 0.70 | 0.065 | 8.88 | 830 | 126 | 3.06 | 1.09 |
| WRFLES-N | 77 | 80 | 9.16 | 0.68 | 0.080 | 8.78 | 661 | 124 | 3.35 | 1.453 |
| WRFLES-U | 77 | 80 | 10.31 | 0.58 | 0.093 | 7.84 | 605 | 120 | 4.34 | 1.99 |
| | 80 | 80 | 8.27 | 0.4 | 0.064 | 2.38 | 1022 | 116 | 1.69 | 0.98 |
| | 80 | 80 | 8.27 | 0.5 | 0.064 | 3.80 | 976 | 111 | 1.87 | 1.09 |
| VPA-TI064 | 80 | 80 | 8.27 | 0.6 | 0.064 | 5.62 | 909 | 106 | 2.05 | 1.2 |
| | 80 | 80 | 8.27 | 0.7 | 0.064 | 8.01 | 843 | 103 | 2.14 | 1.37 |
| | 80 | 80 | 8.27 | 0.8 | 0.064 | 11.18 | 765 | 101 | 2.26 | 1.55 |
| | 80 | 80 | 8.27 | 0.9 | 0.064 | 16.0 | 676 | 98 | 2.36 | 1.79 |
| | 80 | 80 | 8.49 | 0.4 | 0.107 | 2.78 | 904 | 126 | 4.57 | 0.94 |
| | 80 | 80 | 8.49 | 0.5 | 0.107 | 4.25 | 862 | 124 | 4.23 | 1.16 |
| VPA-TI108 | 80 | 80 | 8.49 | 0.6 | 0.107 | 5.96 | 820 | 121 | 3.94 | 1.39 |
| | 80 | 80 | 8.49 | 0.7 | 0.107 | 8.13 | 762 | 118 | 3.84 | 1.57 |
| | 80 | 80 | 8.49 | 0.8 | 0.107 | 11.3 | 677 | 116 | 3.73 | 1.76 |
| | 80 | 80 | 8.49 | 0.9 | 0.107 | 16 | 572 | 111 | 3.58 | 1.94 |

## 2.3 Least-square error fitting procedures

We use the Python least-square error non-linear fitting function in two steps. First, we run the fitting separately for each of the 15 LES cases over the wake region of the various computational domains using Eqs. 8–13, to obtain 15 sets of the five direct-fit parameters $\alpha, \lambda_A, \lambda_W, k_r$, and $\varepsilon_r$ (Table 1). We refer to them as "direct-fit" parameters to distinguish them from those obtained



in the next step. Then, we run the Python fitting again using simple functions of $C_T$ and $TI_\infty$ with three fitting coefficients each, described below, to obtain five functional relationships between the five direct-fit parameters and the two independent variables $C_T$ and $TI_\infty$.

Because it is unpractical to use a different set of fitting parameters for each case and because we want a smooth (i.e., not step-wise) transition between different values of $C_T$ and $TI_\infty$, we want to identify "functional relationships" of the five direct-

fit parameters on a few relevant variables that are turbine- and stability-dependent. There are a few empirical formulations that have been proposed in the literature for $\Delta TI$ and that may be applicable for our purposes. These empirical formulations depend on both $C_T$ and $TI_\infty$ with the following general form (Quarton and Ainslie, 1990; Crespo and Hernández, 1996; Ishihara and Qian, 2018) :

$$\Delta TI \propto a\, C_T^b\, TI_\infty^c. \tag{15}$$

Inspired by these well-established empirical formulations, here we propose that, of the five fitting parameters in Eqs. 9–13 ($\alpha, \lambda_A, \lambda_W, k_r$, and $\varepsilon_r$), three have the form shown in Eq. 15, namely $\alpha, k_r$, and $\varepsilon_r$, while $\lambda_A$ and $\lambda_W$ include an additional dependency on the relevant lengths $D$ and $H$ as follows:

$$\lambda_A = a\, D\, C_T^b\, TI_\infty^c, \tag{16}$$

$$\lambda_W = H + a\, D\, C_T^b\, TI_\infty^c. \tag{17}$$

Intuitively, for a wind turbine with a small diameter, the peak of added TKE occurs at a downstream distance from the tower that is shorter than that for a wind turbine with a large diameter, thus $\lambda_A$ should depend on $D$, as in Eq. 16. Similarly, the TKE peak in the vertical occurs near the rotor top, thus at higher elevations for taller turbines than for shorter ones, which suggests a dependency of $\lambda_W$ on $H$ and $D$, as in Eq. 17.

In summary, the five functional relationships for the five direct-fit parameters are shown in the first row of Table 2. We note that there are now $3\times5 = 15$ functional coefficients $a_i$, $b_i$, and $c_i$, the values of which we obtain via another least-square error fitting. The results are shown in Fig. 2 and Table 2. Although the functional coefficients are empirical in nature, their values provide useful physical hints on the properties of added TKE, as discussed next.

Starting with $\alpha$ (Fig. 2a), its functional relationship is extremely consistent among all the runs and independent of $TI_\infty$,

thus the value of $c$, originally equal to 0.037, is set to zero in Table 2 to simplify the functional relationship and reduce the number of coefficients required overall. The implication is that the magnitude of added TKE in the wake of a wind turbine is essentially independent of atmospheric properties (such as turbulence intensity or stability), but depends only on the turbine operation through its thrust coefficient.

By contrast, $k_r$ (Fig. 2b) is basically independent of $C_T$, thus $b$ is overwritten as zero from the original value of -0.061 in

Table 2. This indicates an interesting finding: that the radial expansion rate of the wake TKE is independent on the turbine operation but is only a function of the amount of background turbulence. This finding is consistent with the literature, as





**Table 2.** Functional relationships for the five direct-fit parameters. The values in parentheses are the original coefficients before manual overwriting to zero.

|  | $\alpha$ | $\lambda_A$ | $\lambda_W$ | $k_r$ | $\varepsilon_r$ |
|---|---|---|---|---|---|
|  | (-) | (m) | (m) | (-) | (-) |
| Equation | $a\,C_T^b\,TI_\infty^c$ | $a\,D\,C_T^b\,TI_\infty^c$ | $H + a\,D\,C_T^b\,TI_\infty^c$ | $a\,C_T^b\,TI_\infty^c$ | $a\,C_T^b\,TI_\infty^c$ |
| a | 0.217 | 3.938 | 1.384 | 0.480 | 0.411 |
| b | 2.269 | -0.472 | -0.429 | 0 (-0.061) | 0.728 |
| c | 0 (0.037) | -0.281 | 0.541 | 1.105 | 0.298 |

**Figure 2.** Performance of the functional relationships at predicting the five direct-fit parameters: a) $\alpha$, b) $k_r$, c) $\lambda_A$, d) $\varepsilon_r$, and e) $\lambda_W$.





$k_r$ has a similar meaning as the well-known expansion rate $k_w$ (also known as just $k$) in the Jensen model(Jensen, 1983), which is typically set to 0.075 onshore, where background turbulence is generally high, and 0.04 offshore, where turbulence is low(Archer et al., 2018). We note that $k_r$ is the expansion of the wake TKE, while $k_w$ is the expansion of the wake wind speed

deficit. As a result of the analysis of the functional relationships for $\alpha$ and $k_r$, the number of fitting coefficients is reduced from 15 to 13.

Next, $\varepsilon_r$ is proportional to both $C_T$ and $TI_\infty$, but the dependency on $C_T$ is stronger ($b$ is more than twice as large as $c$, Table 2). Since $\varepsilon_r$ controls the spread of the added TKE distribution along the annulus of the rotor disk, it is not surprising that its value for the unstable case (WRFLES-U) is under-predicted, while for the stable case (WRFLES-S) it is over-predicted in Fig.

2d.

Focusing on $\lambda_A$ next, since the values of both $b$ and $c$ are negative (Table 2), $\lambda_A$ is inversely proportional to both $C_T$ and $TI_\infty$. However, since both $C_T$ and $TI_\infty$ are always lower than 1 and $b$ and $c$ are both negative and lower than 1, but $c$ is larger in magnitude, the dependency on $TI_\infty$ ends up being dominant. This finding too is physically sound: everything else being the same, the downstream peak of added TKE is expected to be closer to the wind turbine when the background turbulence is

high and further downstream when the background turbulence is low, because high turbulence causes a shorter wake than low turbulence.

By contrast, $\lambda_W$ is inversely proportional to $C_T$ and directly proportional to $TI_\infty$ (Fig. 2e), consistent with negative $b$ and positive $c$ in Table 2. However, due to the additional dependency on $H$ and $D$, the comparison between the lines in Fig. 2e needs to be conducted for the same $D$ and $H$, thus only among lines of the same LES group. For example, looking at the VPA

runs only, a nearly doubling of $TI_\infty$ going from VPA-TI064 to VPA-TI108 causes an about 10-m rise in the vertical placement of the TKE peak, while a nearly doubling of $C_T$ from 0.4 to 0.7 causes less than 5 m of drop in the peak position. Once again, it is physically correct that a more turbulent atmosphere causes a rising of the location of the added TKE peak.

The two-step approach described so far may appear cumbersome, with an initial least-square error fit to the LES data for the five direct-fit parameters and then a second least-square error fit with the functional relationships to finally obtain the values

of the 15 desired functional coefficients. A more straightforward approach would have been to put the functional relationships shown in Table 2 directly into Eqns. 9–13 and then to perform the least-square error fitting to the LES data for the 15 unknown functional coefficients. However, when we tried it, we were unable to reach convergence, possibly because the number of fitting parameters was too high (15 versus 5). We note that Ishihara and Qian (2018)Ishihara and Qian (2018) used a total of 9 fitting parameters.

To assess the performance of the two-step approach, we calculate the root mean square errors (RMSE) over the entire wake regions of all calibration cases (Table 3, table on the left). The proposed analytical formulation outperforms that by Ishihara and Qian (2018) in all cases, as the RMSE of both the direct and the final fit are half as large or lower as theirs on average, and up to 6 times smaller. A general trend that emerges is that the RMSE increases as both $C_T$ and $TI_\infty$ increase. For example, the RMSE of the VPA runs approximately doubles when $TI_\infty$ increases from 0.064 to 0.108 and triples when $C_T$ varies from

0.4 to 0.9. Not surprisingly, the RMSE of the final fit is always higher than that of the direct fit, as, by definition, the direct-fit parameters are those that minimize the error. We note that the RMSEs are relatively large, close to or slightly higher than the





mean value of $\Delta TKE/U_\infty^2$. For example, the RMSE of the final fit for the WRFLES runs is about $1.4 \times 10^{-3}$ and the mean value of $\Delta TKE/U_\infty^2$ for the same runs is about $1.3 \times 10^{-3}$.

**Table 3.** RMSE values of the 15 LES calibration cases (left) and the four validation cases (right) against the benchmark model proposed by Ishihara and Qian (2018), labeled as "IQ2018", the direct fit (with direct-fit parameters from Table 1) and the final fit (with the functional relationships from Table 2). The RMSEs are calculated over a longer domain for the validation cases (0D–20D) than the calibrations cases (up to 10D, see text for details).

| Calibration case | | IQ2018 $(\times 10^{-3})$ | Direct fit $(\times 10^{-3})$ | Fit $(\times 10^{-3})$ | $\overline{\Delta TKE/U_\infty^2}$ $(\times 10^{-3})$ |
|---|---|---|---|---|---|
| WRFLES-S | | 3.77 | 1.19 | 1.37 | 1.22 |
| WRFLES-N | | 3.66 | 1.15 | 1.27 | 1.23 |
| WRFLES-U | | 3.11 | 1.33 | 1.54 | 1.50 |
| VPA-TI064 | $C_T$=0.4 | 1.84 | 0.29 | 0.31 | 0.05 |
| | $C_T$=0.5 | 1.89 | 0.37 | 0.40 | 0.21 |
| | $C_T$=0.6 | 2.29 | 0.55 | 0.59 | 0.43 |
| | $C_T$=0.7 | 3.06 | 0.88 | 0.92 | 0.72 |
| | $C_T$=0.8 | 4.12 | 1.34 | 1.37 | 1.09 |
| | $C_T$=0.9 | 5.40 | 1.98 | 2.02 | 1.61 |
| VPA-TI108 | $C_T$=0.4 | 3.38 | 0.70 | 0.71 | 0.13 |
| | $C_T$=0.5 | 3.16 | 0.78 | 0.79 | 0.33 |
| | $C_T$=0.6 | 3.16 | 0.92 | 0.93 | 0.56 |
| | $C_T$=0.7 | 3.50 | 1.17 | 1.19 | 0.86 |
| | $C_T$=0.8 | 4.22 | 1.54 | 1.56 | 1.24 |
| | $C_T$=0.9 | 5.27 | 2.14 | 2.17 | 1.171 |
| Validation case | | IQ2018 $(\times 10^{-3})$ | Direct fit $(\times 10^{-3})$ | Fit $(\times 10^{-3})$ | $\overline{\Delta TKE/U_\infty^2}$ $(\times 10^{-3})$ |
| XA2017 | | 12.36 | n/a | 3.77 | 2.84 |
| ARC2020 | | 9.54 | n/a | 3.63 | 6.46 |
| SOWFA | | 10.5 | n/a | 2.86 | 4.41 |
| AJU2020 | | 6.67 | n/a | 6.23 | 5.87 |





## 3 Results

### 3.1 Comparison with LES data

First, we look at the individual functions $A, G$, and $W$ along one relevant dimension for the calibration case WRFLES-N. For $\alpha$, the relevant dimension is $x$ and the proposed formulation exhibits the correct features (Fig. 1a): the function is zero at $x_0$, it rapidly increases and then peaks at about 6D, slightly further downstream than indicated by the LES profile, which peaks at about 5D, and then it slowly decreases to nearly zero at 20D. Both the IQ2018 and the CH1996 curves peak near or at the rotor (at $0D$) and retain too much TKE in the far wake after 10D.

The vertical profiles of the two functions $G$ and $W$ (Fig. 1b) also are, for the most part, correctly reproduced by the proposed formulation, including the peak of $\Delta TKE/U_\infty^2$ near the rotor tip and the rapid decrease above it. The second peak of $\Delta TKE/U_\infty^2$ located below hub height is qualitatively reproduced but is too weak; $\Delta TKE/U_\infty^2$ below the rotor decreases without exhibiting the weak negative peak near the surface. The fit by IQ2018 greatly overestimates TKE near the rotor top (by about a factor of 2) and exhibits a spurious peak near the surface.

Next, we compare horizontal and vertical cross-sections of $\Delta TKE/U_\infty^2$ from the LES results, direct fit, and final fit for each of the three stability cases. Starting with a neutral case (VPA-TI064 with CT=0.7) in Fig. 3, the horizontal cross sections at hub height show that the main features and distribution of the direct and final fit for $\Delta TKE/U_\infty^2$ resemble those of the LES results, with a symmetric distribution around the wake axis and two maxima near 6D (slightly further downstream than the LES maxima at $\simeq$ 5D, as mentioned earlier for Fig. 1a). The two maxima collapse into one in the upper part of the rotor (Fig. 3, middle).

In the $x - z$ plane, the $\Delta TKE/U_\infty^2$ maximum is properly located at about 120 m above ground ($z/H \sim 1.5$); the elongated feature of higher $\Delta TKE/U_\infty^2$ extending towards the lower rotor area is more or less captured by the proposed formulation (Fig. 3, right). While the location and magnitude of the maximum, as well as the overall distribution above the rotor, are well captured by both fittings, the wake extent in the x-direction is underestimated and so is the magnitude of $\Delta TKE/U_\infty^2$ below hub height. We note that the magnitude of the maximum $\Delta TKE/U_\infty^2$ is better reproduced by the final fit than by the direct fit.

The comparison between $\Delta TKE/U_\infty^2$ from the LES and the direct and final fits under unstable conditions is also encouraging (Fig. 4 for WRFLES-U). While the direct fit closely mimics the maximum value of $\Delta TKE/U_\infty^2$ under unstable conditions across all three cross sections, the final fit underestimates this term (Fig. 4); the opposite occurred under neutral conditions in Fig. 3. However, the final fit reproduces the $\Delta TKE/U_\infty^2$ propagation along $x$ better than the direct fit.

For stable conditions (Fig. 5), the performance of the proposed fits is more complicated. At hub height, the direct fit underestimates the magnitude of the maximum $\Delta TKE/U_\infty^2$ and the final fit improves it (Fig. 5, left). At the rotor top and in the $z - x$ plane (Fig. 5, middle and right), the opposite happens: the direct fit matches the maximum well, while the final fit underestimates it. It appears that the final fit shifts the maximum of $\Delta TKE/U_\infty^2$ further down in elevation, which causes an increase of TKE around hub height and a reduction near the rotor top.

It is important to note that we did not yet include a treatment of the hub, which causes high $\Delta TKE$ between 1D and 2D in the LES results. The well-known feature that $\Delta TKE/U_\infty^2$ extends further downstream under stable than unstable conditions

**Figure 3.** Cross-sections of $\Delta TKE/U_\infty^2$ under neutral conditions: a) LES (VPA-TI064 with CT=0.7), b) direct fit, and c) final fit in the $x - y$ plane at hub height (left), $x - y$ plane at the rotor top (middle) and in the $x - z$ plane at $y = y_0$ (right).

is correctly captured by our proposed formulation, although the magnitude of the peak is underestimated by the fits in both cases.







**Figure 4.** As in Fig. 3, but for unstable conditions (WRFLES-U).

Lastly, we compare the vertical and horizontal profiles of $\Delta TKE/U_\infty^2$ from the proposed final fit against the formulation of Ishihara and Qian (2018) (IQ2018) and the LES results of citeWuetal2023 under neutral, stable, and unstable conditions at different downstream distances (Fig.6).

In general, IQ2018 overestimates the LES results, especially the magnitude of the $\Delta TKE/U_\infty^2$ peak in the near wake in the vertical (by a factor of 2, Fig.6a) and in the horizontal (by a factor of 3, Fig.6b). By contrast, the proposed fit tends to







**Figure 5.** As in Fig. 3, but for stable conditions (WRFLES-S).

underestimate the maxima in the near wake by up to 30%. In the far wake, the proposed formulation predictions are closely aligned with the LES results; the overestimation by IQ2018 is reduced, but a secondary spurious maximum appears near the surface. Since the LES results include the effect of the hub, while both the final fit and the IQ2018 formulation do not, they both miss the peak in $\Delta TKE/U_\infty^2$ at $x = 2D$ caused by hub (Fig. 3–4, left and right sub-figures).



**Figure 6.** Profiles of $\Delta TKE/U_\infty^2$ from the proposed formulation ("Fit") and from the WRFLES runs (Table 1) along: a) $z$ and b) $y$ at different downstream distances.





### 3.2 Validation

In this section, we compare the performance of the proposed formulation for $\Delta TKE/U_\infty^2$ (i.e., with the final fitting coefficients from Table 2) with data from four other independent studies: the LES studies by Xie and Archer (2017) and by Archer et al. (2020), referred to hereafter as "XA2017" and "ARC2020," respectively; a modified version of the LES by Archer et al. (2013) and Ghaisas et al. (2017), referred to hereafter as "SOWFA;" and the wind tunnel measurements by Aju et al. (2020), referred to as "AJU2020." The four studies are introduced and discussed below and their relevant parameters are listed in Table 4. The

RMSEs for the validation cases are calculated over their respective entire domains starting at 0D to the last point available, which is different in each case (e.g., 10D for SOWFA but just 4D for AJU2020).

| Case | Type | Stability | D | H | $U_{hub}$ | $C_T$ | $TI_\infty$ |
|------|------|-----------|-----|-----|-----------|-------|-------------|
| | | | (m) | (m) | (m s$^{-1}$) | (-) | (-) |
| XA2017 | LES | Unstable | 126 | 87.6 | 8.35 | 0.8 | 0.082 |
| ARC2020 | LES | Neutral | 126 | 90 | 9 | 0.83 | 0.102 |
| SOWFA | LES | Stable | 93 | 80 | 8.3 | 0.8 | 0.074 |
| AJU2020 | Wind tunnel | Neutral | 0.2 | 0.2 | 6.43 | 0.585 | 0.120 |

**Table 4.** Details of the datasets used for the analytical model validation. The label "AX2017" refers to the unstable LES results by Xie and Archer (2017), "ARC2020" to the neutral LES results of Archer et al. (2020), "SOWFA" to a modified version of the LES results by Archer et al. (2013) and by Ghaisas et al. (2017), and "AJU2020" to the neutral wind tunnel experiments of Aju et al. (2020).

XA2017 used the Wind Turbine and Turbulence Simulator (WiTTS) (Xie and Archer, 2015), an in-house flow solver developed at the University of Delaware, for the flow simulation. WiTTS solves the unsteady, filtered three-dimensional Navier-Stokes equations in the incompressible form using the fractional method (Kim and Moin, 1985) and can handle non-neutral

stabilities (Xie and Archer, 2017). The wind turbine, modeled as an actuator line (plus the nacelle), was the REpower 5-MW, with a hub height of 87.6 m and a rotor diameter of 126 m. The simulation was divided into two stages: a precursor stage (without turbines) and a formal stage (with the inclusion of five wind turbines). The five turbines were arranged in a staggered layout with along-wind spacing of approximately 1000 m (which is roughly 8D) and across-wind spacing of roughly 4D. The resolution was 6.25 m in all directions and the surface roughness length was 0.016 m. Only the wake field of the front-row

turbine (labeled "WT1" in their Figure 7) from 2D to 8D is used here, to avoid contamination from the wakes of nearby turbines.

Looking at the results from the unstable run of XIA2017, $\Delta TKE/U_\infty^2$ from the proposed fit matches the LES satisfactorily in the vertical and in the horizontal (Figure 7a and b), with the maximum $\Delta TKE/U_\infty^2$ of the right magnitude and correctly located at the upper part of the rotor tip at 4D and 6D. However, at 8D, the profile along the $z$ direction shows an overestimation

in predicting the maxima near the rotor tip, while correcting reproducing the profile below hub height. The RMSE is larger than that of any of the studies used for calibration, $3.77 \times 10^{-3}$ (Table 3), possibly because the LES included the effect of the nacelle while both analytical formulations do not. The RMSE of the proposed fit is, however, significantly lower than that of




IQ2018 ($12.36 \times 10^{-3}$), which overestimates added TKE at all distances, but especially in the near wake, and introduces, again, an unrealistic peak near the ground.

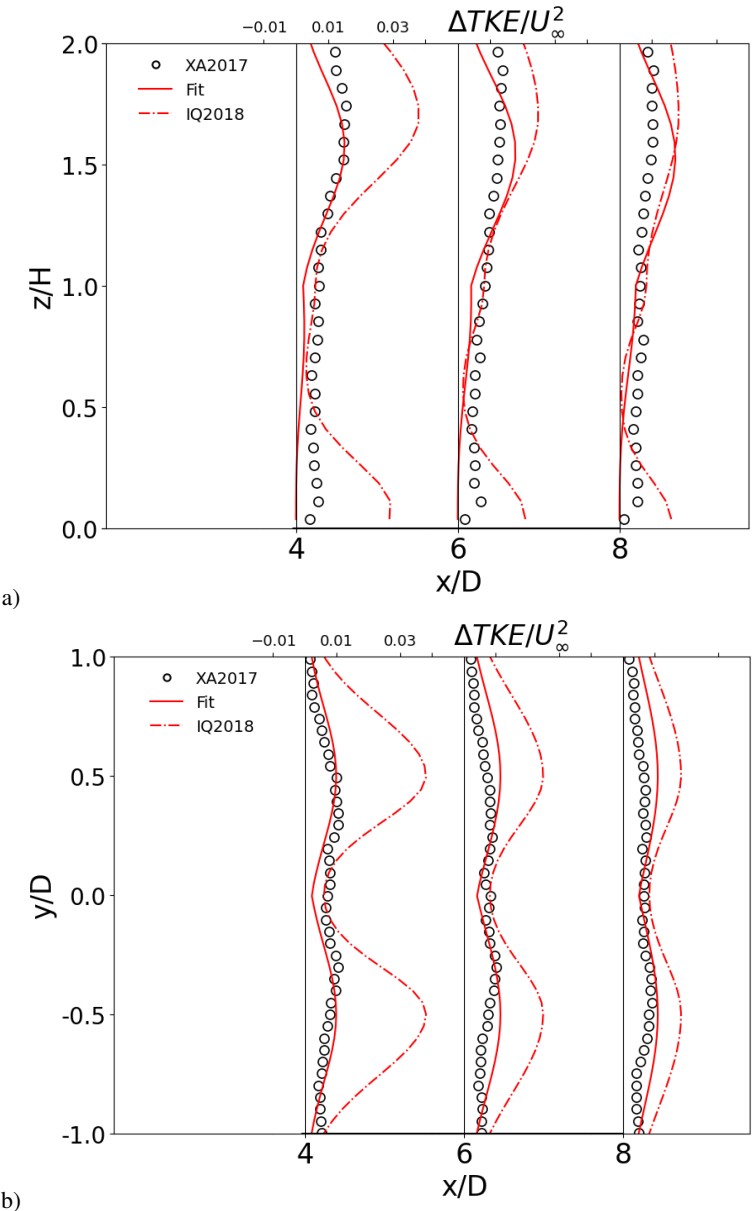

**Figure 7.** Profiles of $\Delta TKE/U_\infty^2$ from the proposed formulation and from the LES data of Xie and Archer (2017), labeled as "Fit" and "XA17" respectively, along: a) the $z$ direction at the centerline and b) the $y$ direction at hub height, taken at different downstream distances for the unstable case.





The second LES study used here for validation is ARC2020 (Archer et al., 2020), in which the LES flow solver Software for Wind Farm Applications (SOWFA) was used with an actuator line model for the wind turbine blades without any treatment for the hub (Churchfield et al., 2012; Archer et al., 2013). The computational domain size was 3000 m × 3000 m × 1020 m with a single wind turbine in the middle and the atmospheric stability was neutral. An idealized NREL 5-MW wind turbine was used with $D = 126$ m and $H = 90$ m. The resolution was set to 200 × 200 × 68 grid points in $x$, $y$, and $z$, respectively,

corresponding to grid cells of approximately 15 m in both horizontal dimensions.

The vertical profiles of $\Delta TKE/U_\infty^2$ from the final fitting match very well those from the LES just after $x = 2D$ (Figure 8a). We note that the slight reduction in TKE near the surface shown in the LES results, which has been mentioned several times in the literature (Archer et al., 2019; Wu and Archer, 2021) and is associated with the reduction in vertical wind shear due to the wind speed deficit, is not reproduced with the proposed fit because it is not accounted for in its equations. Conversely, IQ2018

shows some spurious $\Delta TKE/U_\infty^2$ below the rotor, which is not correct. The y-profiles (Figure 8b) are characterized again by a general underestimation and overestimation of the $\Delta TKE/U_\infty^2$ maxima for proposed fit and for IQ2018, respectively. The proposed fit shows a significantly improved $\Delta TKE/U_\infty^2$ profile compared to IQ2018, providing a more satisfactory prediction that closely matches the LES results starting at $x = 4D$. This validation dataset is well reproduced by the proposed formulation, with an RMSEs of $3.63\times10^{-3}$, about half of the average value of $\Delta TKE/U_\infty^2$ ($6.46 \times10^{-3}$, Table 3, possibly because this

LES study did not include a treatment for the hub.

The third study used for validation, named SOWFA, is a modification of those by Archer et al. (2013) and by Ghaisas et al. (2017), which used the SOWFA solver over a complex mesh of 4000 m × 4000 m × 1000 m with fine refinement (about 3.5 m) in six blocks around up to 48 wind turbines (Siemens 2.3 MW with $D$=93 m and $H$=63.4 m) and coarser (7 m) in the rest of the domain. Various cases were simulated, varying the number of turbines, their layout, and the atmospheric stability. Here we

use an additional stable case, with the same temperature decrease at the bottom boundary of -0.25 K h$^{-1}$ as in Ghaisas et al. (2017), the same layout as "Stg-2SpaX" in Archer et al. (2013), but with westerly flow. To maximize the extent of the wake at fine resolution, we extracted the flow details from the wake of turbine n. 36 (see Fig. 1f in Archer et al. (2013)), approximately 10D in length.

The proposed fitting provides a satisfactory match to the LES profiles, particularly in the z-profiles, in the entire domain,

while a noticeable gap between LES data and IQ2018 still exists in the near wake. We note that the stable LES results show an asymmetry in the y-profiles (Figure 9b), due to the Coriolis force and the resulting Ekman-spiral effect, which is not captured by the proposed formulation and therefore an overestimation by final fitting can be seen in the far wake. Because this LES dataset did not simulate the effect of the hub, the RMSE was relatively low: $2.86\times10^{-3}$, for an average value of $\Delta TKE/U_\infty^2$ of about $4.41\times10^{-3}$ (Table 3, possibly because this LES study also did not include a treatment for the hub.

The fourth study used for validation is AJU2020 (Aju et al., 2020), which describes the Boundary Layer and Subsonic Tunnel (BLAST) experiments that were conducted at the University of Texas at Dallas. The wind tunnel is 30 m long, 2.1 m high, and 2.8 m wide, with cubic blocks of 2.5 cm of height located at 0.2 m between each other on the bottom surface of the test section to achieve a well-developed turbulent boundary layer. Horizontal axis wind turbines, with D = 200 mm and H = 200 mm, were used in the experiments, based on models from Sandia National Labs and locally manufactured at the university.


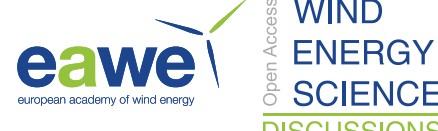

**Figure 8.** Profiles of $\Delta TKE/U_\infty^2$ from the proposed formulation and from the LES data of Archer et al. (2020), labeled as "Fit" and "Archer20" respectively, along: a) the $z$ direction at the centerline and b) the $y$ direction at hub height, taken at different downstream distances under neutral stability.

A particle image velocimetry (PIV) system was used to measure the turbulence along the center axis of each turbine. The tests were conducted in neutral stability and only vertical profiles at selected downstream distances were available. Values near the surface are not reliable due to ground effects.





**Figure 9.** Profiles of $\Delta TKE/U_\infty^2$ from the proposed formulation (labeled "Fit") and from the LES data of Archer et al. (2013) and Ghaisas et al. (2017) (labeled "SOWFA") along: a) the $z$ direction at the centerline and b) the $y$ direction at hub height, taken at different downstream distances under stable conditions.





Comparison of the $\Delta TKE/U_\infty^2$ behind the wind turbine against the experimental data by AJU2020 under neutral conditions indicates that the proposed fit is qualitatively correct but exhibits a large underestimation of the upper rotor peak by over

100% (Figure 10). Not surprisingly, the AJU2020 dataset is associated with the largest RMSE among the four cases used for validation, exceeding $6 \times 10^{-3}$ for an average value of $\Delta TKE/U_\infty^2$ of $5.87 \times 10^{-3}$ (Table 3). This is possibly explained by the fact that this experimental dataset just covers a small region behind the turbine, between $x = 0 - 4D$, where the effect of the hub is more significant. By contrast, this is the only dataset that compares well against the IQ2018 predictions, despite an overestimation of the maximum at 3D and an overall RMSE still larger than that of the proposed fit ($6.67 \times 10^{-3}$, Table 3).

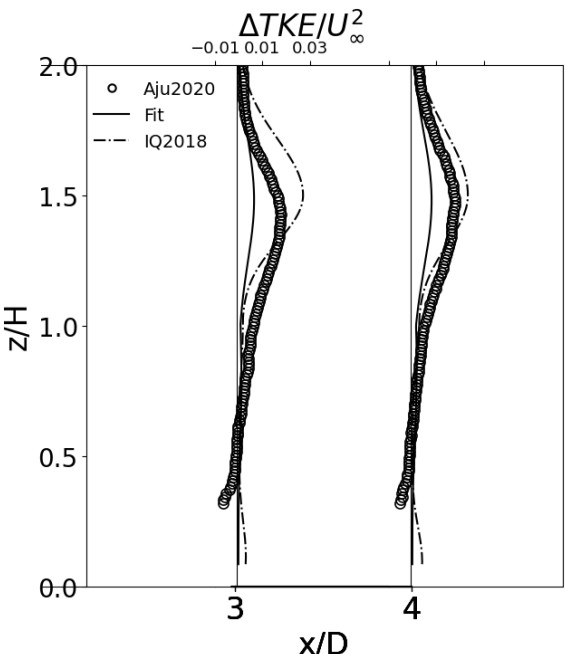

**Figure 10.** Profiles of $\Delta TKE/U_\infty^2$ from the proposed formulation and from the wind tunnel PIV measurements of Aju et al. (2020), labeled as "Fit" and "Aju20" respectively, along the $z$ direction at the centerline under neutral stability.

**4 Conclusions and recommendations**

This study is a first step in addressing the lack of a proper treatment of the turbulence added by wind turbines in current numerical weather prediction models, like the WRF model. An analytical formulation for $\Delta TKE/U_\infty^2$ is presented, comprising five fitting parameters, each with a functional relationship with the thrust coefficient of the turbine, the undisturbed upstream turbulence intensity, the diameter, and the hub height of the wind turbine. The fitting parameters are obtained after a two-step

fitting process based on the LES dataset from a previous study by Wu et al. (2023)), which used the WRF-LES code for three



atmospheric stability cases (stable, neutral, and unstable), and from 15 LES cases in neutral stability with various combinations of $TI_\infty$ and $C_T$ by Vahidi (2024).

The proposed formulation compares well with the LES datasets that were used for the parameter calibration, which is to be expected, with RMSEs of the same order of magnitude as the mean $\Delta TKE/U_\infty^2$, but it is less accurate when compared against
four additional and independent datasets used for validation: an LES study of a 5-MW wind turbine under unstable conditions using the WiTTS code (Xie and Archer, 2017), another LES study of the same 5-MW wind turbine under neutral stability using the SOWFA code Archer et al. (2020), another SOWFA run under stable conditions Archer et al. (2013); Ghaisas et al. (2017), and a wind tunnel experiment with a model wind turbine under neutral conditions Aju et al. (2020).

We conclude that the proposed formulation is promising at predicting the distribution of $\Delta TKE/U_\infty^2$ under all stabilities in
the far-wake, which is the more relevant region for mesoscale studies of the impacts of wind farms on the environment. In the near-wake, the blade geometry, rotor tip, and hub effects have a dominant effect on $\Delta TKE/U_\infty^2$ and the proposed formulation performs worse there than in the far-wake. However, the proposed formulation outperforms that by Ishihara and Qian (2018) in all cases.

The ultimate goal of this research is to eventually insert the $\Delta TKE/U_\infty^2$ formulation, after further improvements to better
capture the near-wake behavior, in numerical weather prediction models to better quantify the possible impacts of wind turbine wakes on the environment. However, in order for the proposed formulation to be effectively used for this purpose, the total $\Delta TKE$ in each grid cell of the mesoscale model needs to be calculated, but the integral of Eq. 8 cannot be obtained analytically. As such, numerical integration is required, which may add a small computational cost to the simulation. In addition, in the presence of multiple wind turbines with multiple overlapping wakes, the issue of superposition of wake-added TKE needs to
be resolved.

*Author contributions.* Archer designed the research, obtained the funding, developed the formulation, and led the research; Khanjari led the Python coding and data processing; Archer and Khanjari analyzed the results and wrote the manuscript; Feroz helped with literature review and formulation development.

*Competing interests.* Archer is a member of the editorial board of Wind Energy Science.

*Acknowledgements.* Partial funding for this research was provided by a grant from First State Marine Wind. The authors are extremely thankful to Dr. Dara Vahidi and Fernando Porté-Agel at the Ecole Polytechnique Fédérale de Lausanne (EPFL, CH) for sharing their LES data and to Dr. Yaqing Jin and his team at University of Texas Dallas (USA) for sharing their wind tunnel data.





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
