# Peer review of "An analytical formulation for turbulence kinetic energy added by wind turbines based on large-eddy simulation"

_Wind Energy Science, 2024_

## Referee Comment (RC3)

Review of the paper "An analytical formulation for turbulent kinetic energy added by wind turbines based on large-eddy simulation" by Khanjari, Feroz, and Archer

The authors propose a new parametrization of the turbulent kinetic energy in the wake of wind turbines. Using LES results and experimental measurements they derive a model based on 5 parameters related to the operational setting of the turbine and spreading of the wake. It is discussed how these parameters are related to the flow physics. The paper is interesting, well written and worth publications.
I have some comments and suggestions that the authors and editor may consider for the final version.

**Major**

- *On line 9, the authors say " The ultimate goal is to insert the proposed formulation, after further improvements, in the WRF model for use within existing or new wind farm parameterizations"*
- The parametrization proposed in the paper captures the variations of the TKE at a scale much smaller than that of WRF (see for example the peaks in Fig.1 b). How this wealth of information can be integrated on coarse grids as those commonly run in WRF (where maybe you have 2-3 points per diameter)?

- I think it would benefit the wind energy community to discuss which turbulent scales you are trying to model. I think the wind turbines add a coherent component which has time scales relatively larger than the incoherent turbulent scales.

- I appreciate a lot the effort of the authors in developing this new parameterization, however, deriving it from a LES which is also dependent on a turbulent model introduces an uncertainty. This process would be perfect if we could run a DNS but of course we can't due to the high Re number. Even if we could run a DNS, the actuator disk model (even the actuator line) may add further uncertainties. To my knowledge, changing the sampling point of the velocity in the actuator model, the spreading (as extensively discussed by Martinez in several papers) or the Smagorinsky constant changes the results. I would recommend the authors to add a paragraph in the final manuscript where they address uncertainties in the simulations used to derive this surrogate.

- A major source of tke is due to the tower and nacelle as shown by Santoni et al. 2017 (Wind Energy) and others. It affects the stability of the hub vortex, the breakup of the tip vortices and the fluxes. Is it irrelevant for the model here proposed, or it could be incorporated through a modified Ct for example? It would be nice if the authors could share some thoughts.

- *Line 275: expansion rate of the wake TKE is independent on the turbine operation but is only a function of the amount of background turbulence.*
  I do not follow this point, if the turbine is operating in off-design conditions it will introduce a lot of turbulence in the wake. This will affect the mixing, fluxes and as a consequence the expansion rate. Maybe this effect I am referring to is taken into account by $\epsilon_r$?

**Minor**

Line 135: the definition of tke is a bit confused. I would suggest saying "where u', v' w' are the fluctuating velocity"... Otherwise, you use them in Eq.5 but define later. There is also a typo in my draft on line133 "andw". I am not sure the overbar is defined. Please check.

Line 153 and $\sigma^2_U \neq \sigma^2_u + \sigma^2_v$. I do not understand this, why they should be related? I do not see the point you are trying to make here.

Line 291 I would also suggest because it increases the mixing, and smooth the peak down

Line 317 I think you refer to A(x) here, because alpha is a constant.

---

## Author Comment (AC1)

**Reply to the reviewers of "An analytical formulation for turbulent kinetic energy added by wind turbines based on large-eddy simulation"**

22 January 2025

Please note that the reviewers' comments are in *italic*, our responses in regular font, and the changes to the manuscript in blue color.

**Reviewer #1**

*The paper presents a new analytical formulation for turbulent kinetic energy in the wake of a single wind turbine. The model proposes a detailed three-dimensional description of the tke field, intended to remain valid in both the near and far wake regions. In total, 15 parameters are introduced and calibrated against the results of large-eddy simulations through a two-step least-squares method. The paper is clear, easy to read, and has the potential to contribute to the improvement of the latest wake models.*

*Major comments:*

- *Line 189, 203: For both Weibull-like laws, the shape parameters are set before starting the fitting process. The authors should motivate the choice of the values kA=2 and kW=4, and clarify why those parameters are not fitted using the two-step least-squares method.*

We really appreciate the comment because we too have actually tried to find the fitted values for $k_A$ and $k_W$. Unfortunately, we could never get convergence with a total of 7 parameters, thus we had to reduce the number of parameters and select reasonable values for $k_A$ and $k_W$ manually. The choice of $k_A = 2$ is motivated by the need for the fitting function $A(x)$ to be exactly zero at $x_0$ (only possible for $k_A > 1$) and to be rapidly increasing past $x_0$, but not too rapidly (which would be the case for $k_A < 2$). For $k_W$, by trial and error we found that a value of 4 would give a steep increase in $TKE$ above the rotor top and a gentler decrease below it, as shown in the LES results. We added the following in the manuscript around line 190:

We set $k_A = 2$ to reduce the overall number of parameters to fit and to obtain a function with the desired properties, i.e., equal to zero at $x_0$ (thus $k_A > 1$) and rapidly increasing past $x_0$, but not too rapidly (which would be the case for $k_A < 2$).

and around line 205:

... the shape parameter $k_W$ is set equal to 4 after a trial-and-error process to ensure a steeper decrease in $\Delta TKE$ above the top tip than below it, as shown in the LES results.

and around line 248:

We attempted to find fitted values also for $k_A$ and $k_W$, but with 7 parameters we could never reach convergence of the least-square error fitting procedure.

- *Line 274: The authors claim that kr is independent of CT. However, Figure 2b shows differences up to 20% between the value of kr at CT=0.4 and at CT=0.9. It would perhaps be interesting to consider an expression other than $CT^b$ for the fitting.*

We would like to clarify that the dotted lines shown in Figure 2b are not those obtained with b=0, but they are the original result of the fitting procedure, with b = -0.061. Since the lines are almost perfectly horizontal, the dependency on $C_T$ is insignificant, which is why we wrote in the manuscript that $k_r$ is basically independent of $C_T$ (now we say that "the fit for $k_r$" is).

We agree that the original points (not the fitted lines) show a weak dependency on the thrust coefficient. However, this dependency is confusing. For the high TI cases (VPA-TI108), it is slightly decreasing with $C_T$, but for low TI cases (VPA-TI064) it is slightly increasing. We believe that this is the reason why the least-square error fit produced a flat line. We added the following around line ...:

By contrast, the fit for $k_r$ (Fig. 2b) is basically independent of $C_T$, despite a weak and conflicting dependency in the direct-fitting values, thus b is overwritten as zero from the original value of -0.061 in Table 2.

As much as we would like to try your suggestion of using a different functional relationship for $C_T$, at this point it would be too massive an effort. Plus the power function has been used in the literature.

- *Figure 2: Because the form of Eq. 15 cannot capture the stability conditions, the differences between the "direct fit" values and that of the functional relationships are often large in the stable and unstable cases. For consistency, shouldn't only neutral conditions be used for calibration?*

In general, stable conditions are characterized by lower $TI$ than unstable conditions, thus the reasoning behind picking $TI$ as the metric for atmospheric conditions was that it would be a decent (although not perfect) proxy for stability. In addition, we want a set of functions that could be used in all stability conditions. If we only calibrated our coefficients on neutral cases, then we would not be able to capture other stabilities at all. By including stable and unstable cases in the calibration, the proposed fitting functions are better equipped to treat non-neutral conditions. Lastly, the stable and unstable cases were clear outliers only for $\varepsilon_r$, as discussed in the manuscript; in the validation, the stable case (SOWFA) was the one with the lowest error (Table 3).

- *Line 305: The authors should clarify what they mean by "entire wake regions" and specify the limits of the regions along y and z as well, as this will influence the value of the RMSE.*

Ali, write the exact limits that you used for the wake regions That's true the size of wake region influence the value of the RMSE, indeed. Thus, in this study, The entire wake region covers an area with y from -1D to +1D and z from 0-2H. Please note that we have different values along the x direction as some LES cases and the experimental test just cover a small regions in downstream

*Minor comments*

- *Line 6: The notation x = 4 – 6D appears a bit confusing. It might be worth considering an alternative notation.*

Changed to 4D–6D.

- *P2: The introduction is rich and well-documented. However, the relevance of the section between lines 28 and 46 is questionable in the scope of this work as it addresses velocity deficit models.*

The discussion was tightened and the two paragraphs were shortened into one, giving a reduction of the number of lines of text from 21 to 13.

- *Line 76: "We note that also Eq. 1 and 2 can be reduced to the same form". This is not true for CT in the case of Eq. 2. The authors should maybe re-phrase this sentence for consistency.*

The sentence was rephrased as follows:

We note that also Eq. 1 can be reduced to this same form and Eq. 2 to a close form (with $(1-\sqrt{1-C_T})^b$ instead of $C_T^b$).

- *Line 133: Typo "u, v, andw"*

  Done.

- *Line 160: Different definitions of $\Delta TI$ exist in the literature. It is worth clarifying which one is used and its connection to $\Delta TKE$.*

  We added the following:

  In particular, the relationship used in this study between added TI ($\Delta TI$) and added TKE ($\Delta TKE$) is:

  $$\Delta TI = \sqrt{\frac{2}{3}} \, \frac{\Delta TKE}{\overline{U}} = \sqrt{\frac{2}{3}} \, \frac{TKE - TKE_\infty}{\overline{U}},$$

  where $TKE_\infty$ is, broadly speaking, the free-stream turbulent kinetic energy. The exact definition of $TKE_\infty$ depends on the type and distribution of the available data. If three-dimensional simulation data are available from a run without turbines (i.e., a precursor run) and a run with turbines, then the point-by-point difference of the time-averaged TKE of the two runs is used to calculate $\Delta TKE$, e.g., for the validation LES datasets described in Section 2.2. If only a simulation with turbines is available, as is the case for the validation LES datasets described in Section 3.2, then the vertical profile of TKE at an upstream distance of $x = x_0 - 2D$ is obtained by calculating at each level the average of TKE over $-3D \leq y - y_0 \leq +3D$, where $x_0, y_0$ are the coordinates of the turbine The value of $TKE_\infty$ to use at each point downstream is, then, the value of TKE in the upstream vertical profile at the same vertical level.

- *Line 189: The expression selected for A(x) is very similar to the one proposed by T Delvaux et al2024 Phys.: Conf. Ser. 2767 092089. The authors could consider providing additional reference for it.*

  Thank you for bringing this article to our attention. We added it to the list of references and cited it in the manuscript around line 192 as follows:

  The Weibull distribution was also recently proposed for the $x$-dependency of added $TI$ by Delvaux et al. (2024, their Eq. 3).

- *Line 194: Missing space after the bracket.*

  Done.

- *Line 277, 279: Missing space before the bracket.*

  Done.

- *Line 351: Typo "citeWuetal2023"*

  Fixed.

- *3.2: The proposed model appears to outperform the model of Ishara and Qian (2018) in most of the validation cases. The comparison could be further enriched with the 3D model of Tian et al. (2022).*

  We added a comparison of the TIAN2022 performance in Figures 5–7 and Table 3.

**Reviewer #2**

*While the authors rightly highlight the need for improved TKE models—since most existing engineering models focus on velocity deficit prediction—the model's reliance on 13 tuning parameters may hinder its broader applicability for diverse case studies. The novelty of this work could be enhanced either by incorporating more advanced ML data-driven models or perhaps adopting a more rigorous physics-based approach. While the authors' efforts in accurate predictions of TKE are appreciated, the current formulation appears more like a straightforward curve fitting to existing datasets rather than a highly novel contribution.*

We agree that the proposed formulation, at the end of the day, is more or less a curve fitting to existing datasets. However, there are two arguments that we would like to offer. The first is that the novelty of this manuscript is not much in the format of the final result (a 13-parameter equation) or how we got there (curve fitting to existing LES data), but rather in having this formulation at all. An analytical model for added TKE by wind turbines simply does not exist in the literature today. One could use the models by IQ2018 or Tian2022 via conversion between TI and TKE, but their models – which also are curve-fitting models – just do not perform well enough. Thus we believe that publishing this formulation in the literature is a necessary first step to then possibly improve it, for example by using more advanced ML methods, as the reviewer suggested. The second argument is that we are not claiming – and are not required by the journal – to give a "highly novel contribution". We are satisfied by giving a valid and useful contribution to the scientific and engineering communities.

- *Title: The use of "analytical" in the title may be misleading. While "analytical" is sometimes used in the literature to broadly describe engineering models, it typically refers to models derived directly from flow governing equations. Given the empirical nature of the developed model, I recommend replacing "analytical" with "engineering" or "empirical" in the title and throughout the text where applicable to avoid confusion.*

  In the wind energy community, the adjective "analytical" is most often used to characterize wake loss models, such as Jensen's, Larsen's, Fradsen's, etc (Frandsen et al., 2006; Bastankhah and Porté-Agel, 2014; Archer et al., 2014, 2018). These models consist of an analytical equation for the wind speed deficit with one or more tuning parameters (e.g., $k_w$ in Jensen's). For example, the Gaussian model for the wind speed deficit, used by Bastankhah and Porté-Agel (2014) among others, is not an analytical solution to any flow governing equations and yet it is called an analytical wake model. Here we do the same, but for added TKE. Thus we prefer to align with the wind energy community tradition and retain "analytical" in the title. In addition, using "engineering" may confuse the readers into thinking that our model is for engineering applications only, which it is not, while "empirical" would suggest that it is based on observations alone, while our model is based also on advanced analytical derivations.

- *Introduction: The discussion on available engineering velocity-deficit models and WRF modelling, while detailed, is not central to this work. These sections could be shortened and presented more concisely to maintain focus.*

  The discussion was tightened and the two paragraphs were shortened into one, giving a reduction of the number of lines of text from 21 to 13.

- *Please clarify how the added TKE and added TI are computed in this work. Especially, this is not trivial when $\sigma_u$ in the wake is less than the one in the incoming flow and thus the added TI becomes negative.*

  We added the following:

  In particular, the relationship used in this study between added TI ($\Delta TI$) and added TKE ($\Delta TKE$) is:
  $$\Delta TI = \sqrt{\frac{2}{3}}\,\frac{\Delta TKE}{\overline{U}} = \sqrt{\frac{2}{3}}\,\frac{TKE - TKE_\infty}{\overline{U}},$$
  where $TKE_\infty$ is, broadly speaking, the free-stream turbulent kinetic energy. The exact definition of $TKE_\infty$ depends on the type and distribution of the available data. If three-dimensional simulation data are available from a run without turbines (i.e., a precursor run) and a run with turbines, then the

point-by-point difference of the time-averaged TKE of the two runs is used to calculate $\Delta TKE$, e.g., for the validation LES datasets described in Section 2.2. If only a simulation with turbines is available, as is the case for the validation LES datasets described in Section 3.2, then the vertical profile of TKE at an upstream distance of $x = x_0 - 2D$ is obtained by calculating at each level the average of TKE over $-3D \leq y - y_0 \leq +3D$, where $x_0, y_0$ are the coordinates of the turbine. The value of $TKE_\infty$ to use at each point downstream is, then, the value of TKE in the upstream vertical profile at the same vertical level.

- *Please improve the quality of all figures, especially figure 2 where Greek symbols are written in Latin letters. This should be avoided and the authors are expected to use proper typesetting to generate figures.*

  We apologize for the sloppy figures. We made a considerable effort to improve the quality of all the figures and made the following modifications:

  - Figure 1: the coordinates were changed to lower case ($x$ and $z$).
  - Figure 2: the titles of the y-axes were changed to Greek letters, where needed, and the subscripts were added; the title of the x-axis was changed from "CT" to "$C_T$"; in the legend, "ti" was replaced with "TI" and a comma was added.
  - Figures 7–10 (now 5–9): replaced "Tian2022" with "TIAN2022", consistent with the rest of the manuscript.
  - Figure 9 (now 5c): the coordinates were changed to lower case ($x$, $y$, and $z$) and the legend item was changed from "Aju2020" to "AJU2020", consistent with the rest of the manuscript.
  - Figure 10 (now 5a): the line across the value 2.0 on the y-axis was removed.
  - Figure 8 (now 6): replaced "Archer2020" with "ARC2020", consistent with the rest of the manuscript.

- *Line 65: The phrase "... flow has less energy" could be misleading, as the far wake typically exhibits more kinetic energy than the near wake due to wake recovery.*

  The sentence was removed.

- *Line 100: The statement, "Notably, Wu et al. (2023) conducted LES that included the effect of atmospheric stability to show that the wind speed deficit behaves differently from the $\Delta TKE$ and that the two are not co-located in the wake region," is somewhat vague. Could you clarify what is meant by "wind speed deficit behaves differently from the $\Delta TKE$"? For instance, does this refer to differences in spatial distribution, magnitude, or temporal evolution? A more precise paraphrasing would enhance clarity.*

  The sentence was modified as follows:

  "the wind speed deficit behaves differently from $\Delta$TKE (e.g., the wind speed deficit reaches the ground within 8D while added TKE remains aloft) and that the two are not co-located in the wake region (e.g., the wind speed deficit peaks at hub height while added TKE near the rotor tip)."

- *Line 125: The statement, "the WRF will add some TKE on its own due to the resolved vertical shear," is unclear. If this detail is not crucial to the discussion or the importance of accurate TKE prediction, consider removing it. Alternatively, if it is essential, please clarify how WRF contributes to TKE through resolved vertical shear and its relevance to the context.*

  This detail is important because it explains why it is crucial not to overestimate TKE in the parameterization. We discussed this issue in detail in the two papers by Ma et al. (2022a,b) cited in the text. The first sentence was expanded to clarify the concept as follows:

  "will add some TKE on its own via the production term in TKE equation, due to the weak, additional, resolved vertical shear caused by the reduced wind speed in the grid cell of the turbines."

- *Line 133: Space is missing in "u,v, andw". Also "and" should not be written in math mode.*

  Done.

- *Line 139: The phrase " ... is the mean wind speed" should likely be "the incoming wind speed".*

  At this point of the discussion, the definition is general, not just related to wind energy applications, thus we really intend to say "mean" and use the overbar on the wind speed symbol ($\bar{U}$). There is no turbine yet into which the flow would be incoming or outgoing.

- *This recently-published paper (Modelling turbulence in axisymmetric wakes: an application to wind turbine wakes, 2024) could be relevant to the discussion provided in this paper, so the authors may want to include it in their literature review.*

  Thank you for suggesting this paper, which we were not aware of and is very relevant. If it had been published earlier, before we submitted our paper, we would have discussed it in detail and added a comparison of the performance of their new analytical model against ours, IQ2018, and TIAN2022. As a compromise, we added a citation to it in reference to the finding that IQ2018 overestimates TKE as follows around line 365:

  "Large overestimates by the IQ2018 model have also been reported recently by Bastankhah et al. (2024) in the near- and far-wake regions."

- *Line 141: "Typically the largest one is $\sigma_u$, followed by $\sigma_v$ (approximately $0.75\sigma_u$ in neutral conditions) and then by $\sigma_w$ (approximately $0.52\sigma_u$ in neutral conditions) (Arya, 2001)." This needs to be moved to the next paragraph after discussing that x is aligned with streamwise direction in this study. Otherwise, this statement is incorrect based on west-east definition for coordinates mentioned initially.*

  The statement is correct and belongs here. In the real atmosphere, on average, the east-west variance is the largest and the other two variances are smaller, as reported. The reviewer is welcome to check the Arya book to confirm that the real atmosphere is not exactly isotropic. We also added a citation to Stull (2017), see his Chapter 18.

- *Line 271: "The implication is that the magnitude of added TKE in the wake of a wind turbine is essentially independent of atmospheric properties (such as turbulence intensity or stability)" . Is that consistent with previous works? Normally, it is expected to observe a negative correlation between the added TKE and the ambient TKE as suggested in Crespo's work and also observed in some numerical and experimental studies.*

- *Figure 2: For some cases such as WRFLES-S or WRFLES-N, there are only one dataset shown in figures. Fitting a line to only one data point sounds tricky. Can you please clarify this?*

  The figure just shows the shape of the functional relationship curves for all the cases, using the $C_T$ and $TI_\infty$ values for each case. For WRFLES-S, for example, we picked the values $C_T$=0.70 and $TI_\infty$=0.065 and plotted the resulting curves in green. The only point shown is the actual original value (sometimes we have many points, like for VPA-TI064). Ideally the point should lay on the curve, or close to it. The farther away it is, the poorer the performance of the fitting. But the actual fitting was done with, literally, millions of points.

- *In several places including but not limited to line 277, there is no space between the text and the parentheses including the citation.*

  Thank you, we fixed it.

- *Line 297: "Once again, it is physically correct that a more turbulent atmosphere causes a rising of the location of the added TKE peak." Can you please explain why is that?*

  The sentence was removed.

- *Line 303: repeated citation!*

  Fixed.

- *Figures 3-5: I'm not sure if contours shown in these figures are really necessary and add value to the paper as both vertical and lateral profiles are provided later. I suggest removing these figures for brevity.*

  We absolutely love these figures and think that they are very important. They are the reason why we chose the Weibull function for the vertical and horizontal directions and the Gaussian function for the radial one. They give a more immediate and intuitive idea of the 3D distribution of added TKE, which is not that well known from the literature, and of the effects of stability on the shape of added TKE. They also allow the readers to appreciate the differences between the first and second fits (smooth) and the real distribution (more jagged).

  But we agree that perhaps showing an example from all three stabilities means too many figures. Thus we kept Figure 3 in the main paper and moved old Figure 4 and 5 to Appendix A (now Figures A1 and A2).

- *All figures especially Figs 6-10 seem to be inconsistent with the main text in terms of font size. They are also too big. Please consider making them smaller and grouping them to make the paper more concise.*

  Figures 6–10 appear to be inconsistent because their datasets are actually inconsistent with one another, since they come from disparate sources. For example, data for x up to 14D were available from ARC2020, but only up to 8D from XA2017. Since we wanted to maximize readability, we made each figure as large as possible. We have now rescaled all these figures to be the same height and combined XA2017 and AJU202 in a single figure (Figure 5).

- *Line 393: "is associated with the reduction in vertical wind shear due to the wind speed deficit, is not reproduced with the proposed fit because it is not accounted for in its equations". The reduced TKE level at lower heights have been reported in several studies, but the current model does not capture that. Please comment on how it can be included in the empirical formulation and how important it is to be modelled.*

  At this early point, it is important for us to avoid the increase in TKE near the ground that plagues IQ2018 and TIAN2022. Reproducing the decreased (or the unchanged) TKE near the ground is a task that we will address in the future. We rephrased the text as follows:

  "We note that the slight reduction in TKE near the surface shown in the LES results, which has been observed and simulated in the literature (Archer et al., 2019; Wu and Archer, 2021), is not reproduced with the proposed fit because it is not accounted for in its equations yet. A way to account for it in the future could be via a correction similar to the $\delta$ function of IQ2018, shown here in Eq. 4."

**Reviewer #3**

*Major points*

- *On line 9, the authors say "The ultimate goal is to insert the proposed formulation, after further improvements, in the WRF model for use within existing or new wind farm parameterizations". The parametrization proposed in the paper captures the variations of the TKE at a scale much smaller than that of WRF (see for example the peaks in Fig.1 b). How this wealth of information can be integrated on coarse grids as those commonly run in WRF (where maybe you have 2-3 points per diameter)?*

  We did not have room in the abstract to expand on this, but the idea is to integrate Eq. 11 over the volume of each grid cell of the mesoscale model in which the wake is present. We expanded on this in the Conclusions:

  "in order for the proposed formulation to be effectively used for this purpose, the total $\Delta TKE$ in each grid cell of the mesoscale model needs to be calculated, but the volume-integral of Eq. 11 cannot be obtained analytically. As such, numerical integration is required, which may add a small computational cost to the simulation."

- *I think it would benefit the wind energy community to discuss which turbulent scales you are trying to model. I think the wind turbines add a coherent component which has time scales relatively larger than the incoherent turbulent scales.*

  We apologize but we are not expert at coherent versus incoherent turbulence and therefore do not feel qualified to entertain such a discussion. But our formulation does not differentiate between the two, it accounts for all the added TKE that has been reproduced with the LES, regardless of coherence.

- *I appreciate a lot the effort of the authors in developing this new parameterization, however, deriving it from a LES which is also dependent on a turbulent model introduces an uncertainty. This process would be perfect if we could run a DNS but of course we can't due to the high Re number. Even if we could run a DNS, the actuator disk model (even the actuator line) may add further uncertainties. To my knowledge, changing the sampling point of the velocity in the actuator model, the spreading (as extensively discussed by Martinez in several papers) or the Smagorinsky constant changes the results. I would recommend the authors to add a paragraph in the final manuscript where they address uncertainties in the simulations used to derive this surrogate.*

  We agree and added the following in the Conclusions:

  "Another limitation of our formulation is that its calibration relies on LES results, which introduce several uncertainties, from the sub-grid turbulent model to the sampling method and the spreading of the actuator line model (Martínez-Tossas et al., 2017). Using only real measurements would not remove all uncertainties either, as measurements have their own intrinsic uncertainties, plus each experiment tends to be specific to the chosen setup and therefore difficult to generalize. Even if we used Direct Numerical Simulation (DNS), which we cannot do yet due to the high Reynolds number of the wind flow, resolving the blades correctly would still require an actuator-line model or similar parameterization, which would add some uncertainty."

- *A major source of tke is due to the tower and nacelle as shown by Santoni et al. 2017 (Wind Energy) and others. It affects the stability of the hub vortex, the breakup of the tip vortices and the fluxes. Is it irrelevant for the model here proposed, or it could be incorporated through a modified Ct for example? It would be nice if the authors could share some thoughts.*

  Thank you for bringing the paper by Santoni et al. (2017) to our attention. We were not aware of it. It appears that Santoni et al. (2017) used a rather "stocky" design for the tower and a very elongated shape for the nacelle, both of which are not realistic for real-size wind turbines. Also, they simulated a wind tunnel case, not a real atmosphere case. Thus it is premature to extend their conclusions to real cases. As such, we think it might be premature to insert a treatment of the nacelle and tower in our formulation. In addition, while a couple of our LES datasets included the effect of the nacelle, none included the effect of the tower, thus we would not be able to calibrate any correction or additional parameter to incorporate their combined effect anyway.

- *Line 275: [radial] expansion rate of the wake TKE is independent on the turbine operation but is only a function of the amount of background turbulence. I do not follow this point, if the turbine is operating in off-design conditions it will introduce a lot of turbulence in the wake. This will affect the mixing, fluxes and as a consequence the expansion rate. Maybe this effect I am referring to is taken into account by $\varepsilon_r$ ?*

  This point follows directly from Figure 2b, where $k_r$ is clearly independent of $C_T$. Recall that we defined $k_r$ as the "radial expansion rate", i.e., the rate of change of the variance of the radial Gaussian curve with $x$, as from Eq. 15:

  $$\sigma_r(x) = k_r\,(x - x_0) + \varepsilon_r D,$$

  and

  $$k_r = \frac{\partial \sigma_r}{\partial x}.$$

  We did not consider any off-design conditions because we had no data for such cases.

- *Line 135: the definition of tke is a bit confused. I would suggest saying "where u', v' w' are the fluctuating velocity"... Otherwise, you use them in Eq.5 but define later. There is also a typo in my draft on line133 "andw". I am not sure the overbar is defined. Please check.*

  The definition was changed as follows:

  "where a bar $\overline{(\cdot)}$ indicates a mean and a prime $(\cdot)'$ refers to a fluctuating component, i.e., the difference between the instantaneous and the mean wind component, e.g., $u' = u - \bar{u}$"

- *Line153 and $\sigma_U^2 \neq \sigma_u^2 + \sigma_v^2$. I do not understand this, why they should be related? I do not see the point you are trying to make here.*

  The sentence in parenthesis was removed.

- *Line 291 I would also suggest because it increases the mixing, and smooth the peak down*

  The sentence was modified as follows:

  "because high turbulence causes a shorter wake than low turbulence, increases mixing, and smooths down the peak."

- *Line 317 I think you refer to A(x) here, because alpha is a constant.*

  Correct, thank you for catching the typo, which is now fixed.

**References**

Archer, C., Vasel-Be-Hagh, A., Yan, C., Wu, S., Pan, Y., Brodie, J., and Maguire, A.: Review and evaluation of wake loss models for wind energy applications, Applied Energy, 226, 1187–1207, https://doi.org/10.1016/j.apenergy.2018.05.085, 2018.

Archer, C. L., Colle, B. A., Monache, L. D., Dvorak, M. J., Lundquist, J., Bailey, B. H., Beaucage, P., Churchfield, M. J., Fitch, A. C., Kosovic, B., Lee, S., Moriarty, P. J., Simao, H., Stevens, R. J. A. M., Veron, D., and Zack, J.: Meteorology for Coastal/Offshore Wind Energy in the United States: Recommendations and Research Needs for the Next 10 Years, Bulletin of the American Meteorological Society, 95, 515 – 519, https://doi.org/10.1175/BAMS-D-13-00108.1, 2014.

Archer, C. L., Wu, S., Vasel-Be-Hagh, A., Brodie, J. F., Delgado, R., Pé, A. S., Oncley, S., and Semmer, S.: The VERTEX field campaign: Observations of near-ground effects of wind turbine wakes, Journal of Turbulence, 20, 64–92, https://doi.org/10.1080/14685248.2019.1572161, 2019.

Bastankhah, M. and Porté-Agel, F.: A new analytical model for wind-turbine wakes, Renewable Energy, 70, 116–123, https://doi.org/10.1016/j.renene.2014.01.002, 2014.

Bastankhah, M., Zunder, J. K., Hydon, P. E., Deebank, C., and Placidi, M.: Modelling turbulence in axisymmetric wakes: an application to wind turbine wakes, Journal of Fluid Mechanics, 1000, A2, https://doi.org/10.1017/jfm.2024.664, 2024.

Frandsen, S., Barthelmie, R., Pryor, S., Rathmann, O., Larsen, S., Højstrup, J., and Thøgersen, M.: Analytical modelling of wind speed deficit in large offshore wind farms, Wind Energy, 9, 39–53, https://doi.org/10.1002/we.189, 2006.

Martínez-Tossas, L. A., Churchfield, M. J., and Meneveau, C.: Optimal smoothing length scale for actuator line models of wind turbine blades based on Gaussian body force distribution, Wind Energy, 20, 1083–1096, https://doi.org/10.1002/we.2081, 2017.

Wu, S. and Archer, C. L.: Near-ground effects of wind turbines: Observations and physical mechanisms, Monthly Weather Review, 149, 879 – 898, https://doi.org/10.1175/MWR-D-20-0186.1, 2021.

---

## Author Response (AR3)

**Second reply to the reviewers of "An analytical formulation for turbulence kinetic energy added by wind turbines based on large-eddy simulation"**

18 February 2025

Please note that the reviewers' comments are in *italic*, our responses in regular font, and the changes to the manuscript in blue color.

**Reviewer #1**

- *In the author's response to reviewers, the relationship between $\Delta TI$ and $\Delta TKE$ has been added. However, I believe there is a typo in it, i.e. $\sqrt{\Delta TKE}$ instead of $\Delta TKE$?*

  Thank you so much for catching the typo, which is now corrected.

**Reviewer #2**

- *Regarding my comments on the curve-fitting nature of this work, I did not mean to suggest that the developed model is not useful. My intention was to provide a suggestion for enhancing the scientific quality of the study. However, I understand that this is ongoing research and indeed an important step toward more accurate TKE prediction.*

  Thank you for the clarification.

- *Regarding my comment on using "empirical" instead of "analytical": I still do not fully agree with your response. While the example you provided (i.e., the Gaussian wake velocity model) incorporates empirical assumptions (e.g., wake shape), it simplifies governing flow equations to predict the maximum velocity deficit, so there is a difference here. That said, I appreciate that the terminology is somewhat subjective, and I respect your choice to use "analytical" instead of "empirical."*

  Thank you for accepting our use of "analytical".

- *Regarding my comment on line 141 (based on the original manuscript), I understand your point that turbulence in the ABL is not isotropic, and the streamwise component is typically larger than the other two. However, while the streamwise direction aligns with the wind, it is not necessarily west-east. The current wording suggests that x is aligned with the west-east direction, regardless of wind direction. This is why I suggested discussing the values of $\sigma_u$, ... in the next paragraph, where you later clarify that in your work x coincides with the streamwise direction.*

  There is often a confusion about the coordinates to use in the wind energy community. The IEC uses the convention of aligning the $x$ axis with the streamwise direction, thus with the average wind direction. With such a convention, the variance of wind speed $\sigma_U^2$ basically coincides with that of the u component $\sigma_u^2$. But in meteorology, the $x$ axis is not the streamwise direction, it is always the west-east direction. In such a case, the variance of wind speed is not well approximated by that of the u component only. I (Archer) have a paper in review in WES that talks exactly about these issues (Archer, 2024). In this manuscript, the story we are telling is:

1. If turbulence was isotropic, the three variances would be identical;

2. The "real" atmosphere, which is studied in meteorology and atmospheric science, is not isotropic because the west-east direction, which is conventionally the x direction, is more turbulent than the other two;

3. In wind energy science and engineering, the x axis is conventionally aligned with the mean wind, or, in other words, the streamline direction. Even with this convention, the turbulence is not isotropic because the standard deviation along x is largest.

We added a small clarification to the manuscript and a more accurate citation:

"the IEC standard recognizes that the three standard deviations in Eq. 7, even with the convention of aligning the $x$-axis streamwise, should be different from one another and recommends that any wind velocity field for turbulence models used for standard turbine classes satisfy the following conditions: $\sigma_v \geq 0.7\sigma_u$ and $\sigma_w \geq 0.5\sigma_u$ (International Electrotechnical Commission, 2019, Table C.1, Eq. C.10)

- *Regarding my comment on line 271, I did not see a response. This may have been due to an error in document generation. While it is not a major issue, I wanted to mention it in case you intended to provide clarification that did not appear in the final version.*

We apologize for missing your comment in our previous reply. Here is our reply:

"This finding is consistent with several published studies. Crespo and Hernández (1996) proposed that the maximum added TI depends only on $C_T$ in the near wake and has a very weak dependency on $TI_\infty$ in the far wake (i.e., the power exponent is -0.0325, see Eq. 2); the equations for maximum added TI proposed by both Larsen et al. (1996) and Frandsen (2007) depend only on $C_T$; and the formulations for maximum added TI by IQ2018, shown in Eq. 3, and by TIAN2022, shown in Eq. 5, have a much weaker dependency on $TI_\infty$ than on $C_T$. However, other studies have proposed that $TI_\infty$ has a non-minor role (Ainslie, 1988; Xie and Archer, 2015)."

**References**

Archer, C. L.: Brief communication: A note on the variance of wind speed and turbulence intensity, Wind Energy Science Discussions, 2024, 1–8, https://doi.org/10.5194/wes-2024-159, 2024.